health and disease and epidemiology

COVID-19, attitude change, pandemic, effective reproduction

**Author for correspondence:**
Ilan Fischer
e-mail: ifischer@psy.haifa.ac.il

# The behavioural challenge of the COVID-19 pandemic: indirect measurements and personalized attitude changing treatments (IMPACT)

Ilan Fischer[1], Shacked Avrashi[1], Tomer Oz[1],
Rabab Fadul[1], Koral Gutman[1], Daniel Rubenstein[2],
Gregory Kroliczak[3], Sebastian Goerg[4]
and Andreas Glöckner[5]

[1]School of Psychological Sciences, University of Haifa, Haifa, Israel
[2]Department of Ecology and Evolutionary Biology, Princeton University, Princeton, NJ, USA
[3]Faculty of Psychology and Cognitive Science, Adam Mickiewicz University, Poznan, Poland
[4]Department of Biotechnology and Sustainability, School of Management, Technical University of Munich, Munich, Germany
[5]Social Cognition Center Cologne, University of Cologne, Cologne, Germany

IF, 0000-0002-2622-7743; DR, 0000-0001-9049-5219;
GK, 0000-0001-6121-0536; SG, 0000-0002-1740-6870;
AG, 0000-0002-7766-4791

Following the outbreak of COVID-19 pandemic, governments around the globe coerced their citizens to adhere to preventive health behaviours, aiming to reduce the *effective* reproduction numbers of the virus. Driven by game theoretic considerations and inspired by the work of US National Research Council's Committee on Food Habits (1943) during WWII, and the post-WWII Yale Communication Research Program, the present research shows how to achieve enhanced adherence to health regulations without coercion. To this aim, we combine three elements: (i) indirect measurements, (ii) personalized interventions, and (iii) attitude changing treatments (IMPACT). We find that a cluster of short interventions, such as elaboration on possible consequences, induction of cognitive dissonance, addressing next of kin and similar others and receiving advice following severity judgements, improves individuals' health-preserving attitudes. We propose extending the use of IMPACT under closure periods and during the resumption of social and

economic activities under COVID-19 pandemic, since efficient and lasting adherence should rely on personal attitudes rather than on coercion alone. Finally, we point to the opportunity of international cooperation generated by the pandemic.

## 1. Introduction

The outbreak of the COVID-19 pandemic made governments around the world to coerce their citizens to comply with preventive health behaviours. These actions are expected to reduce the *effective* reproduction number, $R$. While the *basic* reproduction number, $R_0$, reflects the contagiousness or transmissibility of an infectious agent [1], $R$ reflects the expected number of additional cases that one infected individual will generate, given the effectiveness of implemented health and social interventions. In other words, the more successful health interventions are, the lower the value of $R$, and the less people will move from the state of being susceptible to the state of being infected, within a specific period of time. If $R < 1$, the disease will gradually die out [2–4].

To reduce the *effective* reproductive number of virus infections, governments enforced numerous emergency regulations. Individuals were asked to adhere to constraining behaviours, such as staying at home, keeping social distance, repeatedly washing hands and avoiding meeting seniors. Although restricting personal freedom, these behaviours generate health benefits for both the individual and the entire population. They are expected to lower contact probabilities with vulnerable populations, especially seniors or those with immune deficiencies; hence preventing hospitals from being overrun, flattening the infection curve, lowering its peaks and contributing to the lowering of $R$ and the burn-out of the disease.

Potential interactions among individuals, who may either be infected or susceptible to infection, form a social dilemma. Each individual benefits from others' cooperation, yet is motivated to violate the restrictions himself [5]. However, a closer look at the case of COVID-19 shows that it is actually not a typical social dilemma. An individual that assumes others do not adhere to health regulations is himself motivated to better and more strictly adhere to these regulations, as both the risks of not adhering and the advantages of adhering become more critical for one's own well-being. In fact, the scenario is better modelled by the Chicken game ([6–8]; table 1), a game that motivates the players to cooperate, even when assuming the opponent does not. While previous studies have indeed demonstrated the emergence of cooperation in Chicken games, individuals are *not* expected to perceive the COVID-19 pandemic as a strategic interaction or analyse its game theoretic characteristics. Nonetheless, modelling social interactions under COVID-19 as Chicken games highlights two important aspects: (i) it suggests that the behavioural challenge of reducing the effective reproductive number $R$ matches the motivations of the involved individuals. Hence, individuals' behaviour is not only the problem, but, given appropriate interventions, may actually become the solution. (ii) Since the Chicken game is a member of a category of games termed similarity-sensitive games (SSGs), cooperative behaviour may be induced simply by raising the perception of similarity with other players [9–11].

To design appropriate health-promoting interventions under COVID-19, we address not only the cooperative nature of the Chicken game. We also aim to influence attitudes and risk perceptions, which have been shown to influence health behaviours in general [12,13] as well as specifically in the case of infectious diseases [14,15]. Moreover, interventions need to diagnose health attitudes and risk perceptions, identify individuals' weak spots and offer an assortment of behavioural tools that have the potential to induce adherence to health-promoting regulations. Ideally, these initiatives should neither be coerced, nor interfere with individuals' free will, as shown by the recurring protests against lockdowns imposed by many states in the USA and other countries. While there is a considerable amount of research measuring risk perceptions concerning COVID-19 (e.g. [16–21]), it does not involve direct psychological interventions on the level of individuals. This is the issue we address in our research.

To construct non-coercive behavioural intervention measures, we draw on the work of the US National Research Council's Committee on Food Habits, conducted eight decades ago during WWII [22]. Motivated by food shortages of several products in the USA and Europe, the committee investigated why people eat, what they eat and then practised several methods for changing these habits. Like the Committee on Food Habits, the present research examines individual attitudes towards several health regulations and applies an assortment of psychological principles that enable

**Table 1.** COVID-19 as a Chicken game played between two players, with ordinal pay-offs, that express the rank-order of the outcomes, ranging from best (4) to worst (1). The left value in each cell is the pay-off obtained by player A and the right value is the pay-off obtained by player B. The four pay-offs comprise: unilateral other's adherence (4)—the best outcome since the other player is keeping the regulations, while the player himself is not keeping the regulations, wherein he or she is free from personal constraints while still being protected by health regulations kept by the other player; mutual adherence (3)—both players are constrained by keeping the regulations, but both minimize the likelihood of contracting the virus; unilateral own-adherence (2)—being constrained by keeping health regulations while the other player is not. This is indeed an imbalanced and unfair outcome. Nonetheless, knowing the other player does not adhere makes self-adherence even more valuable (as the alternative is to switch to no adherence and obtain a lower pay-off); mutual no adherence (1)—if both sides do not keep health regulations, the individual may benefit from not being constrained, yet suffer from the highest likelihood of contracting the virus. Note that the choice to adhere is the Maxi-min strategy of the game—the strategy that provides the better outcome while comparing the two minimal pay-offs, one for each strategy. Since the game is symmetric, the intersection of the two Maxi-min strategies (i.e. mutual cooperation) may be regarded as the natural outcome of the game [8]. Importantly, the game is also a similarity-sensitive game (SSG) [9,10]; hence, the higher the perception of similarity with the opponent the more likely one is to adhere to health regulations.

|  |  | B | |
|---|---|---|---|
|  |  | adherence | no adherence |
| A | adherence | 3, 3 | 2, 4 |
|  | no adherence | 4, 2 | 1, 1 |

re-examination and improvement of these attitudes. Unlike the committee, we do not go beyond attitudes. We keep strict anonymity of the participants and do not monitor their succeeding behaviours. Research has shown that frequent exposure and experience with an attitude-related object increases the accessibility of the attitude, which in turn raises the likelihood that it will induce consistent behaviour [23]. Moreover, attitudes are both unique and specific. They are exclusive to each individual [24] and specific to each topic or behaviour. The more specific an attitude, the more likely it is to predict behaviour [25].

While many studies have addressed the link between persuasion, attitudes and behaviour, we focus here on an assortment of studies that revealed simple and potent effects that can easily be adjusted to the behavioural challenges imposed by the COVID-19 pandemic. We also extend the search to relevant effects of decision-making, altruism and cooperation. The main criteria for the inclusion of these processes are their established efficacy and the simplicity of their administration and adaptation to a computerized survey disseminated via the Internet. We neither assume that the following list of effects is complete nor that it represents an optimal choice.

The present study is first of all an intervention project that aims to benefit the participants by helping them to understand and develop internally motivated adherence. Being constrained by the goal of providing immediate benefits, we do not use a bottom-up methodology that studies isolated effects and progresses by testing ever more complex interactions. Instead, we apply a cluster of psychological effects, embedded in a short computerized survey that enables testing the overall efficacy of the entire bundle of attitudes. Initially, we assigned a few participants to a simple repetition condition, but once sufficient evidence had been collected, we continued by assigning all participants to the full intervention condition.

## 1.1. Descriptions of behavioural effects embedded in the study

*Loss aversion* is a crucial aspect of prospect theory [26]. It assumes that the utility function, which assigns subjective utilities to objective values, is concave for gains, convex for losses and steeper for losses than gains. This results in loss aversion; that is, losses are weighed more than for gains of equal size. To benefit from this effect, we frame all numerical COVID-19-related questions in terms of *infected people* (losses) rather than in terms of *healthy people* (gains).

*The Yale Attitude Change Approach* suggests that the effectiveness of persuasive communication depends, among others, on the source of the message [27] and the prestige of the medium on which the message is communicated with [28]. As we approach the wider public (and not the medical

or scientific community), an appropriate source of information should be widely acknowledged and at the same time be regarded as trustworthy. Hence, we selected a Wikipedia paragraph that cites the World Health Organization as a source for the description of the COVID-19 pandemic. We also added several laboratory and university logos on the header of the survey to boost prestige and credibility.

*The Elaboration likelihood model* [29] suggests that the more a person actively thinks and processes the content of a message, the more likely he or she is to accept the content of the message. To benefit from this effect, respondents are asked to read text, answer numerical questions and compose their own persuasive messages.

*Cognitive dissonance theory* [30] suggests that individuals are motivated to reduce dissonance between inconsistent cognitions. According to the theory, when people behave inconsistently with their attitude and cannot find external justification for their behaviour, they experience dissonance that threatens their self-image. To restore congruency and eliminate the dissonance, people may either change attitudes or adjust their behaviour. Which aspect is more likely to change depends on the level of commitment and strength of each element. To benefit from this effect, respondents are asked to explicitly express their personal health recommendations in writing. Having expressed two clear and somewhat irrevocable recommendations, they are expected to align their health-preserving behaviour with the written recommendations, consequently reducing cognitive dissonance and strengthening health attitudes.

The study of *Advice taking* under the context of virus infections [31] has shown that participants increase advice taking by 20% when perceiving the outbreak of the virus as being severe. However, telling apart severe from mild outbreaks required developing expertise that was gained over several repeated trials. To benefit from the increase in advice taking, we first consolidate severity perceptions of the COVID-19 pandemic. To this aim, we primed the severity of the situation by asking respondents to rate the severity of the situation, on a scale, ranging from 'Not good/Serious' to 'Extremely severe'. Immediately after the assessment, they receive advice in the form of an abridged set of the recommended health regulations (i.e. do not leave your home, avoid meeting senior citizens, maintain social distance from other people, make sure to wash hands and maintain strict hygiene).

*Kin selection* suggests that an organism favours the reproductive success of his relatives. Formalized by Hamilton [32], the benefit of an altruistic act towards a recipient is weighed with the relatedness to the recipient and should exceed the costs incurred by the altruistic act. To benefit from this effect, we ask respondents to consider how to implement health regulations in order to protect their family and relatives.

*Subjective expected relative similarity* (SERS) shows that cooperation in SSGs such as the Chicken game does not only depend on the expected pay-offs, but also on the extent of perceived strategic similarity with the other party [9,10]. In short, SERS predicts that individuals are more willing to engage in cooperative behaviour if they have a high similarity perception of the people they interact with. To benefit from this effect, we ask respondents to consider how to apply health regulations in order to protect others who are similar to themselves.

The *bandwagon effect* shows that the probability of an individual to adopt an idea increases with respect to the proportion of others who have already done so. As more people believe in something, others are more likely to join [33]. Although the survey does not measure the impact of this effect, we mention *the group of people who adhere to health authorities' regulations* in the closing remarks of the survey.

The *foot in the door* technique is expected to induce compliance by asking for a small request followed by a bigger one. The compliance to the small request is expected to increase the likelihood of the compliance to the bigger subsequent request [34,35]. Although the survey does not measure the impact of this technique, we thank the participants for taking part in the survey (i.e. complying with a small request) and ask them to continue adhering to health authority regulations (i.e. a bigger request).

## 2. Method

Participants from various countries were approached via email and social networks, and asked to participate in a short survey by clicking on a specified link. Dissemination was channelled through academic and societal influencers, using personal requests, professional organizations, Facebook, Instagram and WhatsApp. Everyone who received the survey was also asked to continue and distribute it. Questionnaires and appeals were written in English, Hebrew, Arabic, Polish and German, as well as in Spanish, Dutch and French. Due to the yet insufficient numbers of responses in some of the languages, we report language-specific statistics only for the first five languages.

The introductory message read as follows:

As part of the worldwide struggle against the spread of the coronavirus, we develop behavioral interventions that help researchers and policymakers to improve the communication with the public and assist in the distribution of efficient COVID-19 related health instructions. We need volunteers to fill out and continue distributing short questionnaires, which takes no longer than 10 minutes to complete. This project is run by the laboratory for the study of Social Dilemmas at the University of Haifa, in collaboration with researchers from Princeton University, Adam Mickiewicz University, University of Cologne, and Technical University of Munich.

Respondents that followed the link were first provided with a clear declaration of their rights as participants, including voluntary non-obligated participation, the right to refuse or to discontinue participation at any moment, confidentiality of the data and strict anonymity of participants' identity. About 20% of those viewing this page confirmed a mandatory consent statement and went on to actually participate in the survey.

To indirectly measure initial, implicit, health attitudes, participants answered two questions regarding each of four health regulations. The first question asked how many individuals, out of 1000, that keep a specified instruction are likely to *be infected with the Coronavirus*. The second question asked how many individuals, out of 1000, that *do not* keep the specified instruction are likely to *be infected with the Coronavirus*.

Unlike most surveys that take a direct interest in the actual values provided by the respondents, our approach does not require that participants provide correct or meaningful estimates that correspond with risks in the real world. In fact, we calculate the differences between the estimates within each pair of questions (i.e. the number of people who are likely to be infected if they adhere to the specified instruction, and the number of people who are likely to be infected if they do not adhere to the same specified instruction). This difference serves as an indirect measure of respondents' attitudes. A small difference indicates disbelief in the efficiency of the specified health instruction addressed within the question pair, and vice versa. It is also important noting that taking into account the lack of actual knowledge, participants might rely on various heuristics for their probability judgements [36]. They might anchor their judgements on available numbers, a phenomenon termed the anchoring and adjustment heuristic. Individuals might use the retrievability of instances (e.g. from the media) to assess these numbers (availability heuristic). Furthermore, the absolute level of perceived infection risk probabilities might be influenced by factors such as experienced dread and general knowledge about the risk [18,37]. Importantly, as both the adherence and non-adherence questions are elicited on the same scale, the difference between the responses is not expected to be biased itself by these factors.

The questions addressed: leaving home for essential needs only or leaving home as usual, maintaining or not maintaining a social distance, keeping or not keeping hygiene (specifically washing their hands, avoiding touching their faces and various public surfaces), or being 65 years of age or older and meeting or avoiding relatives and acquaintances. The questions were presented in different orders. Motivated by prospect theory [26], all questions were framed as losses (i.e. contracting the virus rather than staying healthy). A person who *does not believe* in the efficacy of a specific health instruction is likely to respond by providing similar numbers with only a small difference separating the two estimates of becoming infected by adhering or not adhering to suggested governmental guidelines, and vice versa. A person believing in the efficacy of the instruction is likely to provide different and more distinctive estimates for each of the two questions. After all questions were answered, the software identified the weakest attitude of each participant (the attitude with the smallest difference) for later use in two specific tasks. At this stage, participants were shown a paragraph from Wikipedia that briefly described the COVID-19 pandemic, its effects, spread and its recognition by the World Health Organization as a pandemic. This paragraph not only provided basic information, but also pointed to the credibility of its source, as proposed by the Yale attitude change approach (see also appendix A).

The 2019–20 COVID-19 pandemic is an ongoing pandemic of coronavirus disease 2019 (COVID-19), caused by severe acute respiratory syndrome coronavirus 2 (SARS-CoV-2). The outbreak was first identified in Wuhan, Hubei, China, in December 2019, and was recognized as a pandemic by the World Health Organization (WHO) on 11 March 2020.

The virus is typically spread during close contact and via respiratory droplets produced when people cough or sneeze. Respiratory droplets may be produced during breathing but the virus is not considered airborne. It may also spread when one touches a contaminated surface and then their face. It is most contagious when people are symptomatic, although spread may be possible before symptoms appear. The time between exposure and symptom onset is typically around five days, but may range from two to fourteen days.

The survey then addressed the weakest attitude of each respondent (calculating the difference between adherence and non-adherence responses for each health regulation, and choosing the attitude with the smallest difference) and asked him or her to provide a written answer for why this specific measure is important for stopping the spread of the Coronavirus. After completion, the following request was to compose a short sentence that could be used in the media to explain his or her reason. In this way, people who did not adhere to health regulations would experience a cognitive dissonance between their low regard for the effectiveness of this measure and the fact that they are having to account for its importance, which they could then resolve by adopting health-preserving attitudes and follow with actual adherence. These items also provided the opportunity for cognitive elaboration, as proposed by the elaboration likelihood model. The following question then made references to both *kin selection* and to the theory of *SERS*, asking 'How do you propose to protect *your family, your relatives* and *people similar to you*?', hence addressing both the next of kin and similar others. As both questions generate behavioural expressions in the form of written paragraphs, they also provide a form of enacted behaviour that should motivate the adjustment of attitudes in accord with cognitive dissonance theory. From this point onwards, we did not restrict participants' responses to specific attitudes. To further activate the increased likelihood of advice taking in the case of severe virus outbreaks [31], we primed participants' severity perceptions, asking them to rate the severity of the pandemic by choosing one of the following rankings: Not good/serious, Extremely serious, Severe, Extremely severe. They were then told that 'since the outbreak of the virus is truly dangerous, health authorities have recommended the following guidelines, among others: Do not leave your home and avoid meeting senior citizens; Maintain a distance of at least 2 meters from other people (note that the exact distance was adjusted to the recommendation valid for each country); Make sure to wash hands and maintain strict hygiene'. These abridged guidelines were then followed by a request to explain 'Which of these health authority guidelines is especially important to your family, relatives and people similar to you', again addressing kin selection and SERS. Finally, the participants were presented with a second set of the initial numeric evaluations for the number of infected people, those who adhere to the regulations and those who do not, for each of the four main health regulations.

The final passage, although not measured any more, was also phrased in accordance with two persuasion techniques, the bandwagon and the foot in the door effects. We thanked the participants for 'joining all those who have participated in this survey and are adhering to health authority guidelines', and also thanked them for their time and efforts in filling the questionnaire (i.e. complying to a small request) consequently asking them 'to continue following the guidelines of the health authorities' (i.e. a bigger request).

# 3. Results and analyses

## 3.1. Dissemination dates and languages

The link was clicked by 22 887 people, and 4361 participated and completed various versions of the survey. After running several pilot surveys and optimizing content and software, we analysed intervention and repetition group responses provided by 3102 participants (65% females, average age 38.89 years, s.d. = 16.76). Responses were provided in the following languages: 1214 answered in English, 728 in Hebrew, 497 in Arabic, 466 in Polish, 126 in German and a few more responded in Dutch and Spanish. About half of English language responses were collected in the USA; Hebrew and Arabic responses were collected mainly in Israel. Pilot studies were run between 24 and 27 March 2020. The data of the repetition group were collected from 25 March to 2 April, and the data for the intervention group were collected between 28 March and 18 April 2020.

The intervention group received the full interventions cluster reported above (intervention condition), whereas the repetition group only repeated their probability assessments a second time (repetition condition) without additional interventions in between.

Hereafter, we report results associated with four differences, illustrated in figure 1: (i) the difference between adherence and non-adherence in the first set of questions; (ii) the difference between adherence and non-adherence in the second set of questions; (iii) the shift in the number of estimated infected people who *adhere* to health regulations between the first and second set of questions; (iv) the shift in the number of estimated infected people who *do not adhere* between the first and second set of questions.

Figure 2 depicts rank-ordered differences between adherence and non-adherence in the first set of questions for four tested health attitudes (difference 1 in figure 1), revealing participant initial least

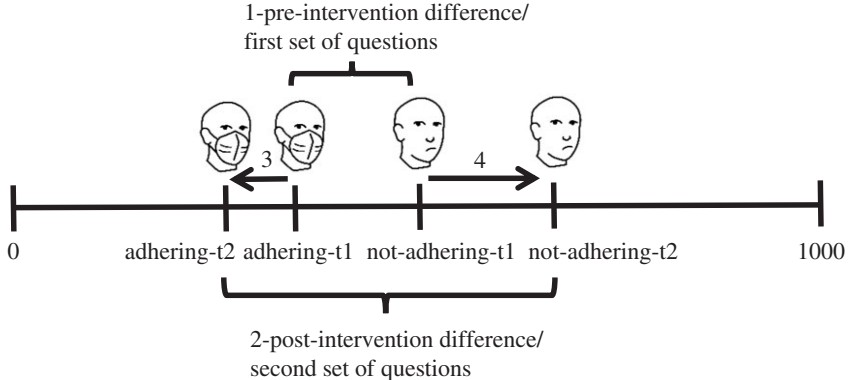

**Figure 1.** Depiction of the dependent variables and hypotheses for both the intervention and repetition condition. Initially, participants' attitude reflects a small difference between the number of people who will be infected if they adhere or if they do not adhere to health regulations. After the intervention (or within the second set of questions), the attitudes are expected to change, reflecting a lower number of infected people for those who adhere, and a higher, asymmetric, change for those who do not adhere to health regulations. (1) The difference between adherence and non-adherence in the first set of questions. (2) The difference between adherence and non-adherence in the second set of questions. (3) The shift in the number of estimated infected people who adhere to health regulations between the first (t1) and second (t2) set of questions. (4) The shift in the number of estimated infected people who do not adhere between the first (t1) and second (t2) set of questions. Table 2 shows the shift % for '1' and '2' differences. Table 3 shows shift % for '3' and '4' differences.

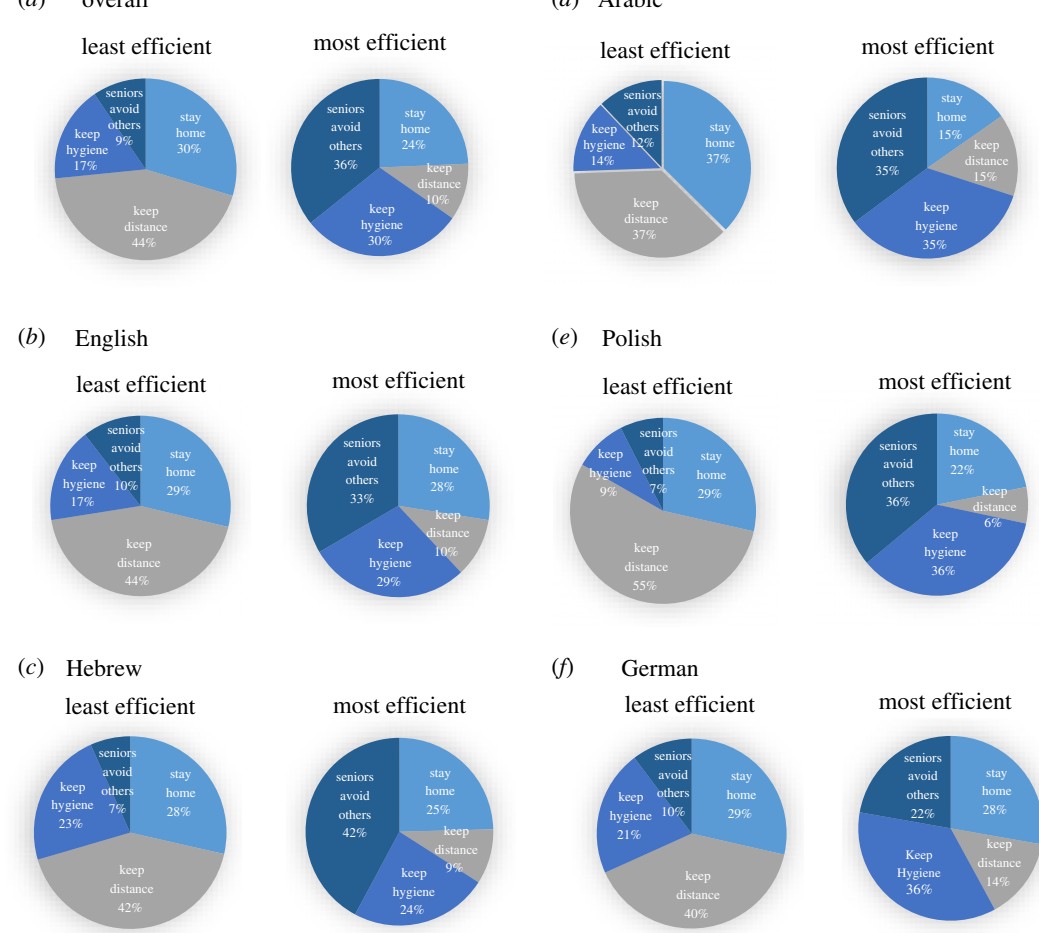

**Figure 2.** Across languages (*a*, *n* = 3102) pre-intervention perceptions of the least and most efficient health regulation, followed by: English-speaking participants (*b*, *n* = 1214), Hebrew-speaking participants (*c*, *n* = 728), Arabic-speaking participants (*d*, *n* = 497), Polish-speaking participants (*e*, *n* = 466) and German-speaking participants (*f*, *n* = 126).

**Table 2.** Comparison of *differences* in the estimations of infected people between those who *do not adhere* and those who *adhere*, calculated twice: for the first set of questions (Diff-pre) and for the second set of questions (Diff-post). The % change is calculated by: $100 \times$ (Diff-post − Diff-pre)/Diff-pre. The same indices are provided separately for the repetition and intervention conditions within various language samples. Differences are calculated from the indices appearing in appendix B.

| repetition | | | intervention | | |
|---|---|---|---|---|---|
| Diff-pre | Diff-post | % change | Diff-pre | Diff-post | % change |
| all languages | | | | | |
| 138.12 | 234.99 | 70.13 | 149.37 | 265.96 | 78.05 |
| English | | | | | |
| 100.67 | 189.48 | 88.22 | 137.74 | 248.66 | 80.53 |
| Hebrew | | | | | |
| 124.33 | 206.96 | 66.46 | 135.61 | 244.59 | 80.36 |
| Arabic | | | | | |
| 159.46 | 278.01 | 74.34 | 172.80 | 334.89 | 93.80 |
| Polish | | | | | |
| 195.37 | 266.57 | 36.44 | 170.16 | 275.08 | 61.66 |
| German | | | | | |
| — | — | — | 164.83 | 273.48 | 65.92 |

and most efficient health attitudes. Table 2 then shows the differences between 'the difference between adherence and non-adherence in the first set of questions' (difference 1 in figure 1) and 'the difference between adherence and non-adherence in the second set of questions' (difference 2 in figure 1). These shifts reflect the efficacy of the two conditions in enhancing participants' health attitudes. Figure 3 depicts the shift in the number of estimated infected people who *adhere* to health regulations between the first and the second set of questions (difference 3 in figure 1) and the shift in the number of estimated infected people who *do not adhere* between the first and second set of questions (difference 4 in figure 1). These data allow estimating the efficacy of both tracks, adhering and not adhering, in changing health attitudes after being informed about the seriousness of the virus and ways to stop its spread. Table 3 further shows these differences as shift percentages. Appendix B lists basic means, differences and ANOVA statistics. The full dataset collected for this study is available as an electronic supplementary material, and may also be obtained by email from the corresponding author.

## 3.2. Initial health attitudes

Here, we describe the indirect measurements of pre-intervention attitudes in our samples with respect to the four health regulations: staying at home, keeping a social distance, keeping hygiene and seniors avoiding contact with the relatives and acquaintances. To indirectly measure attitudes, we examine the set of numerical questions presented at the beginning of the survey, across and within languages. Perceived efficiency of each attitude is estimated by subtracting the number of people that are believed to contract the virus (out of a theoretical group of 1000 people) if they *adhere* to a specific instruction from the number of people believed to contract the virus (out of a theoretical group of 1000 people) if they *do not adhere* to the instruction. The smaller the gap the *less effective* the specific health instruction is perceived by a person, and vice versa. This indirect approach attempts to minimize socially desirable answers, the answers one gives because he knows they are appreciated and regarded as being correct. The questions we used are provided in appendix A.

After calculating these indirect measures, we rank-order the perceptions to determine the least and most efficient instruction for each participant. Figure 2 shows that overall ratings across subject pools from different cultures and linguistic backgrounds are quite similar. The detailed descriptive statistics are provided in appendix B. In our full sample, 44% of the participants regarded 'keeping social

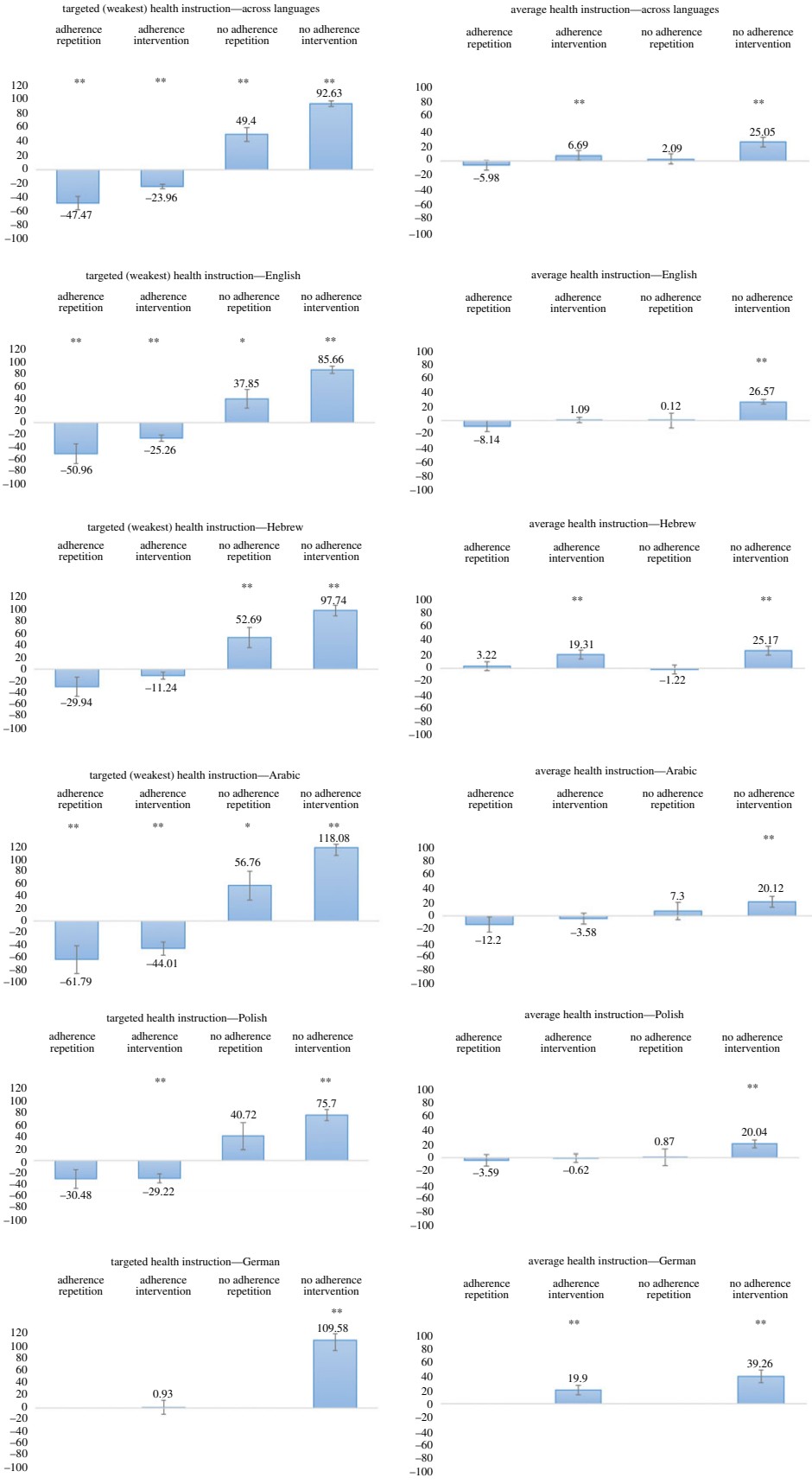

**Figure 3.** (*Caption overleaf.*)

**Figure 3.** (*Overleaf.*) Attitude change efficiencies with standard errors. The left panels show the changes in the perception of participants' estimates for the targeted attitude of each individual, i.e. the weakest personal pre-intervention attitude. This attitude is calculated by subtracting the estimated number of people who will contract the COVID-19 virus (out of 1000 people) if they adhere to a specific instruction, from the number of people who will contract the virus if they do not adhere to the same health instruction. After calculating the differences for all four health instructions, the attitude with the smallest difference, hence the attitude perceived as least efficient, is selected as the weakest attitude for each individual participant. The changes presented in the figure are calculated as the difference between pre- and post-intervention estimates, separately calculated for the numbers provided for the adherence and non-adherence questions. A successful intervention should show negative values for adherence (indicating that the post-intervention attitude reflects a reduced number of people who are likely to contract the virus) and positive values for non-adherence (indicating that the post-intervention attitude reflects a rise in the number of people who are likely to contract the virus). Each figure shows the same statistics for the repetition (only repeated estimates) and the full intervention conditions (comprising all practised attitude change procedures). The right panels show parallel non-personalized (non-targeted) indices, averaged across four health regulations. The upper two panels present cross-sample indices, lower panels present samples collected in five languages. The German sample was run only with the intervention treatment. $^{*}p < 0.05$, $^{**}p < 0.01$. Appendix B lists all respective ANOVA tests and effect sizes.

distance' as least efficient, 30% regarded 'staying at home' as least efficient, 17% 'keeping hygiene instructions' and only 9% regarded 'seniors avoid meeting relatives' as the least efficient instruction. By contrast, 'seniors avoid meeting relatives' was perceived by 36% of the participants as the most efficient measure for seniors not to be infected, followed by 30% for 'keeping hygiene', 24% for 'staying at home' and 10% for 'keeping social distance' all with respect to the probability of average people being infected.

'Keeping social distance' was regarded as the least effective instruction in all languages (English 44%, Hebrew 42%, Arabic 37% tied with stay at home, Polish 55%, German 40%). English- and Hebrew-speaking participants regarded 'seniors avoid meeting others' as the most efficient instruction (33 and 42%). In Arabic and Polish, 'seniors avoid meeting others' was tied with 'keep hygiene' (35 and 36%) and among German-speaking participants 'keep hygiene' was the instruction that was regarded the most effective (36%).

## 3.3. Intervention and repetition efficacy

Next, we tested the effect of the intervention. For each participant, we diagnosed the weakest personal instruction, which is defined as the health instruction for which the estimated likelihood of contracting the virus in the case of adherence is the most similar to the case of non-adherence. Therefore, it is the health instruction most likely to be ignored by the participant.

Figure 3 depicts the means and statistical effect sizes for both the weakest/targeted attitude (left panels) and across all four health perspectives (right panels). The panels on the left show the changes in the perception of participants' estimates for the personally targeted attitude of each individual (differences 3 and 4 in figure 1). Each panel shows the statistics for the *repetition* and the *intervention* condition averaged for adherence and non-adherence attitudes. The panels on the right show parallel non-personalized (non-targeted) indices, averaged across four health regulations. The upper two panels present cross-sample indices and the lower panels present samples collected in five languages. All figures depict meaningful improvement in health-preserving attitudes, highlighting the advantages of the intervention and the higher efficacy of the non-adherence route. Table 2 shows the percentage shift from 'difference 1' to 'difference 2' (illustrated in figure 1). Over and across language samples, we obtained meaningful shifts of expanding differences between pre- and post-intervention, or between the first and second set of questions, ranging from 36 to 94% in both conditions. This shows that learning more about the disease, purposely reasoning about its prevention and answering repeated questions helped participants to develop improved health-preserving attitudes (table 2). Examining the adherence and non-adherence attitude changes (differences 3 and 4 in figure 1), depicted in figure 3 and reported as per cent shift in table 3, shows that while both adherence and non-adherence attitudes change over time, the most effective changes are driven by concerns of *non-adherence* in the *intervention* group, revealing a shift of 26% across language samples (while the repetition group revealed only a shift of 13%). However, in the less effective *adherence* measure, the *intervention* group revealed a shift of −11%, while the *repetition* group revealed an even better shift of

**Table 3.** Percentage changes in respondents' estimates of (i) infected people who *adhere* to the targeted health regulation and (ii) infected people who *do not adhere* to the targeted health regulation between the first and the second set of questions, calculated for the repetition and intervention conditions and various language samples. Negative values for adherence indicate participants' attitude shift, expressing the understanding that less people will be infected if they adhere, while positive values for *no adherence* indicate participants' attitude shift, expressing the understanding that more people will be infected if they do not adhere to health regulations. Shifts are calculated from the indices listed in appendix B.

| | all languages | | English | | Hebrew | | Arabic | | Polish | |
|---|---|---|---|---|---|---|---|---|---|---|
| | repetition | intervention | repetition | intervention | repetition | intervention | repetition | intervention | repetition | intervention |
| adherence | −19.05 | −11.41 | −20.62 | −12.20 | −13.14 | −5.85 | −22.70 | −21.40 | −13.40 | −11.47 |
| no adherence | 12.76 | 25.78 | 10.88 | 24.84 | 14.96 | 29.82 | 13.15 | 31.20 | 9.63 | 17.82 |

−19%. Testing the interaction between time (first and second set of questions) and condition (repetition or intervention) separately for *adherence* and *non-adherence* attitudes reveals significant differences between groups for both ($F_{1,3100} = 7.83$, $p < 0.01$ for the *former* and $F_{1,3100} = 18.17$, $p < 0.01$ for the *latter*). Note that attitude improvements are reflected by positive shifts for non-adherence (i.e. more people will be infected following no adherence) and negative shifts for adherence (i.e. less people will be infected following adherence). Also note that adherence is associated with a single negative aspect, as the judgements assess the number of *infected* people, while non-adherence is associated with a *twofold* negative framing, as it relates to the number of *infected* people following *non-adherence* to health regulations.

## 4. Discussion

The present study was motivated by the social challenge imposed by the COVID-19 pandemic, namely the reduction in the effective reproductive number *R*. Analysis of the structure of the social dilemma underlying the situation, specifically the choice between adherence and non-adherence, revealed a typical pay-off structure of a Chicken game. This showed that *in theory*, individuals' adherence to health regulations is congruent with the public goal of reducing the effective reproductive number *R*. Since the Chicken game is a SSG, it also motivated one of the interventions, namely the addressing of similarity with the opponent as a means towards increased cooperation and enhanced adherence to health regulations.

We then took an action research approach, aiming to implement a behavioural intervention that enhances health attitudes and reduces the likelihood of individuals being infected with COVID-19. Three important aspects of the present study are: (i) **I**ndirect **M**easurements, (ii) **P**ersonalized interventions, and (iii) **A**ttitude **C**hanging **T**reatments (IMPACT). To achieve indirect attitude measurements, health attitudes were *not* inferred from isolated responses, but from the differences among responses within question pairs, referring once to the number of people that are expected to be infected if they *adhere* to a specific health regulation, and once to the number of people that are expected to be infected if they *do not adhere* to a specific health regulation. The smaller the gap, the lower the efficiency of the specific regulation is regarded. After eliciting initial health attitudes, we applied nine well-known effects in a single and relatively short intervention, and also ran a plain repetition condition. We repeated the attitude assessment process, allowing to test the effectiveness of the intervention cluster. Once sufficient evidence had been accumulated from the repetition condition, we ceased running it and assigned all additional participants to the intervention condition.

Pre-intervention attitudes reveal the overall low weight assigned to social distancing and the much greater assumed efficiency of seniors not being infected when visited by relatives and acquaintances. Although figure 2 shows some differences between subject pools, the main patterns are similar across various language samples. We assume that these indirectly obtained ratings best reflect what people are assuming to be meaningful measures, which should also correlate with their behaviour.

The low weight assigned to social distancing comes as a surprise, as the best way to prevent illness is to avoid being exposed to the virus and exposing others to the virus. This is not the case for any of the samples. Hence, forthcoming economic recovery under COVID-19 that requires renewing employment and staffing of work places should strongly emphasize the critical role of social distancing. As shown in the present work, providing information about the disease and its prevention and motivating deliberate processing of the information helps to attain this goal.

The instruction most people perceived as being most efficient is that of seniors avoiding meeting relatives and acquaintances. There are many explanations for the high perceived efficiency of the seniors avoiding meeting others measure. This finding is in line with rational concerns as well as classic findings and models of risk perception. Particularly, seniors are most vulnerable to be infected with COVID-19. Social quarantine measures (i.e. them not being visited by relatives and acquaintances) can, therefore, be assumed most helpful for them to decrease their infection risk on rational grounds. Furthermore, given that fatality rates are particularly high for the seniors, COVID-19 is particularly dreadful for them. Dread is the core determinant for risk perceptions in lay people [37]; therefore, this finding is in line with previous work on risk perception. Furthermore, the estimated effect might have been particularly strong since only this measure did not concern all people themselves, but only a subgroup. Most of the participants did not belong to the group of seniors (90% were below the age of 65), which might have contributed to the larger effect, since many people were not directly affected by this measure (see also [38]).

Overall, we found that participants in the repetition and the intervention condition improved their *combined* adherence and non-adherence attitudes, increasing the difference between first and second estimates. The improvements ranged from 36% to 94% across conditions and language samples, suggesting that those populations who may have had less information in regard of COVID-19 benefited more than others from their participation in the survey. Singling out the best way to improve health attitudes shows that a twofold negative framing approach (how many people who *do not adhere* to health regulations will be *infected*) and the application of the entire attitude change intervention induced an average shift of 26%.

Due to the use of a cluster of several effects, we are unable to point to strengths and weaknesses of specific effects or interactions. Such a study would require a bottom-up approach that does not coincide with the goals of providing health benefits and developing applicable methods in the shortest time possible. Clearly, some principles and interactions may be more potent than others in changing COVID-19 health attitudes. Moreover, many other psychological interventions may also contribute to achieving enhanced attitude changes. Among them, the arousal of a moderate amount of fear that motivates people to analyse the information more carefully [39,40], the focus on independence for individualistic cultures and interdependence for collectivistic cultures [41,42], or the emphasis on injunctive norms that describe what other people approve or disapprove [43,44].

It is also important to extend the study to repeated application of the IMPACT process, thus attaining stronger and longer lasting effects. Assuming that attitude changes follow a typical learning curve, we may expect to see meaningful changes during the initial application phase, which are then moderated and gradually approach an asymptote. Future research should also examine ways to increase the attractiveness of the IMPACT process and improve its dissemination. We recommend embedding IMPACT in school and higher education curricula, integrate it in social agendas of both governmental and private organizations and provide participants with various benefits in award for their participation. Clearly, possible mutations of the virus and emerging health and economic developments require constant monitoring of the situation and the undertaking of timely adjustments and improvements of behavioural interventions. For example, the wearing of face masks has not yet been recommended while we initiated the described interventions, but has since become a key factor in the prevention of virus infections. As countries around the globe attempt to restore social and economic activities, more emphasis should be put on enhancing individual responsibility and adherence to health regulations. As shown by the present study, this goal may be attained by: Indirect Measurements, Personalized interventions and Attitude Changing Treatments.

Finally, we would like to address a somewhat hidden spillover effect, which is not reflected in the responses we received, but in the origins of these responses. Addressing the pandemic, we obtained responses from people residing in 77 different countries. We obtained responses from: Albania, Argentina, Bahrain, Bangladesh, Belgium, Botswana, Chile, Colombia, Denmark, Egypt, El Salvador, Estonia, India, Indonesia, Morocco, New Zealand, Oman, Qatar, Syria, Turkey, Uganda and many more. Clearly, the motivation to cooperate was as widespread as the pandemic itself. Returning to the description of the behavioural challenge imposed by COVID-19 as a Chicken game where cooperation is motivated by fear that the other party is not likely to cooperate (i.e. a Maxi-min strategy) and by the perception that the other party is sufficiently similar, hence likely to choose the same alternative (as explained by SERS; [9,10,45]), we suggest that COVID-19 has the power to motivate cooperation. Both the fear of the consequences of non-cooperation by the other party and the amplified perception of similarity (as both parties are threatened by the same pandemic) motivate people to cooperate. This spillover effect should not be overlooked, as it provides a unique opportunity for resolving lasting disputes and international conflicts.

Ethics. Two separate IRBs have been issued at the University of Haifa, 091/20 and Princeton University-12760.

Data accessibility. All data are available in the electronic supplementary material, titled 'The behavioral challenge of COVID-19-DATA.xlxs'.

Authors' contributions. I.F. initiated the project, supervised its development, participated in data analysis, wrote the first draft and revised the manuscript. S.A. and T.O. participated in the project's development and in data collection, prepared software, maintained data, participated in data analysis and manuscript revisions. R.F. contributed to the creation of the Arabic version, participated in the project's development, data collection and recruited participants. K.G. participated in the project's development, data collection and recruited participants. D.R. participated in the development of the study, contributed to the creation of the English version of the survey and the drafting of the manuscript. G.K. contributed to the creation of the Polish version of the survey, recruited participants and contributed to drafting of the manuscript. S.G. and A.G. contributed to the creation of the German version of the survey, recruited participants and contributed to drafting of the manuscript. All authors gave their approval for publication of the manuscript.

Competing interests. All authors declare having no competing interests.

Funding. We received no funding for this study.

Acknowledgements. We gratefully acknowledge the assistance of Dan Ariely, David Budescu, Tamar Kugler and Bawa Jain in the dissemination of the survey; Michal Klichowski in recruiting the Polish sample of participants; Ines Taylor, Terence Das Dores Cruz, Larisa Olteanu, Dorota Burda-Fischer and Lior Savranevsi in preparing multi-language versions; Adam Fischer in preparing the graphic illustration; and the anonymous reviewers of the manuscript for their helpful comments.

# Appendix A

*Description of the questions used as dependent measures in both sets of questions*

Out of 1000 people who do not leave their homes at all, how many do you think will contract the Coronavirus? (This measure was not used in the analyses, instead we used the next, less strict question.)

Out of 1000 people who leave their homes for essential needs only, how many do you think will contract the Coronavirus?

Out of 1000 people who leave their homes as usual, how many do you think will contract the Coronavirus?

Out of 1000 people who do not maintain a distance of 2 m (6.5 feet) from other people, how many do you think will contract the Coronavirus? (Distances were adjusted to specific regulations of each country.)

Out of 1000 people who do maintain a distance of 2 m (6.5 feet) from other people, how many do you think will contract the Coronavirus? (Distances were adjusted to specific regulations of each country.)

Out of 1000 people who do not take care to wash their hands, and do not avoid touching their faces and various surfaces (such as handles, doors, elevator buttons, public surfaces and more), how many do you think will contract the Coronavirus?

Out of 1000 people who do make sure to wash their hands and avoid contact with their faces and other surfaces (such as handles, doors, elevator buttons, public surfaces and more), how many do you think will contract the Coronavirus?

Out of 1000 seniors (aged 65+) who continue to meet with relatives and acquaintances, how many do you think will contract the Coronavirus?

Out of 1000 seniors (aged 65+) who avoid meeting with relatives and acquaintances, how many do you think will contract the Coronavirus?

*Wikipedia texts included in English* (also translated to German, Polish, Dutch, Spanish and French) surveys (2019–20 coronavirus pandemic, 23 March 2020, paras 1 and 2), https://en.wikipedia.org/wiki/2019%E2%80%9320_coronavirus_pandemic.

'The 2019–20 COVID-19 pandemic is an ongoing pandemic of coronavirus disease 2019 (COVID-19), caused by severe acute respiratory syndrome coronavirus 2 (SARS-CoV-2). The outbreak was first identified in Wuhan, Hubei, China, in December 2019, and was recognized as a pandemic by the World Health Organization (WHO) on 11 March 2020.

'The virus is typically spread during close contact and via respiratory droplets produced when people cough or sneeze. Respiratory droplets may be produced during breathing but the virus is not considered airborne. It may also spread when one touches a contaminated surface and then their face. It is most contagious when people are symptomatic, although spread may be possible before symptoms appear. The time between exposure and symptom onset is typically around five days, but may range from two to fourteen days'.

*Hebrew and translated Arabic versions* were retrieved from:

https://he.wikipedia.org/wiki/%D7%9E%D7%92%D7%A4%D7%AA_%D7%94%D7%A7%D7%95%D7%A8%D7%95%D7%A0%D7%94_(2019%E2%80%932020)

# Appendix B

| language | condition | dependent variable | time | mean | s.d. | mean dif. | n | F | p-value | $\eta_p^2$ |
|---|---|---|---|---|---|---|---|---|---|---|
| all | repetition | targeted adherence | pre | 249.14 | 283 | −47.47 | 471 | 24.905 | 0.000 | 0.05 |
| | | | post | 201.67 | 229.06 | | | | | |
| all | intervention | targeted adherence | pre | 210 | 248.82 | −23.96 | 2631 | 58.958 | 0.000 | 0.022 |
| | | | post | 186.04 | 219.66 | | | | | |
| all | repetition | targeted no adherence | pre | 387.26 | 324.52 | 49.4 | 471 | 24.997 | 0.000 | 0.05 |
| | | | post | 436.66 | 316.14 | | | | | |
| all | intervention | targeted no adherence | pre | 359.37 | 311.57 | 92.63 | 2631 | 561.36 | 0.000 | 0.176 |
| | | | post | 452 | 324.12 | | | | | |
| all | repetition | average adherence | pre | 175.9 | 181.72 | −5.98 | 471 | 1.73 | 0.189 | 0.004 |
| | | | post | 169.92 | 175.4 | | | | | |
| all | intervention | average adherence | pre | 127.88 | 146.16 | 6.69 | 1124 | 9.302 | 0.002 | 0.008 |
| | | | post | 134.57 | 150.26 | | | | | |
| all | repetition | average no adherence | pre | 494.34 | 288.57 | 2.09 | 471 | 0.145 | 0.703 | 0 |
| | | | post | 496.43 | 294.39 | | | | | |
| all | intervention | average no adherence | pre | 469.45 | 297.44 | 25.05 | 2631 | 119.19 | 0.000 | 0.043 |
| | | | post | 494.5 | 306.94 | | | | | |
| English | repetition | targeted adherence | pre | 247.11 | 274.05 | −50.96 | 114 | 10.084 | 0.002 | 0.082 |
| | | | post | 196.15 | 213.27 | | | | | |
| English | intervention | targeted adherence | pre | 207.09 | 244.75 | −25.26 | 1100 | 27.225 | 0.000 | 0.024 |
| | | | post | 181.83 | 212.65 | | | | | |
| English | repetition | targeted no adherence | pre | 347.78 | 294.19 | 37.85 | 114 | 6.159 | 0.015 | 0.052 |
| | | | post | 385.63 | 291.56 | | | | | |

(*Continued.*)

(Continued.)

| language | condition | dependent variable | time | mean | s.d. | mean dif. | n | F | p-value | $\eta_p^2$ |
|---|---|---|---|---|---|---|---|---|---|---|
| English | intervention | targeted no adherence | pre | 344.83 | 302.9 | 85.66 | 1100 | 228.56 | 0.000 | 0.172 |
| | | | post | 430.49 | 316.01 | | | | | |
| English | repetition | average adherence | pre | 181.31 | 190.08 | −8.14 | 114 | 1.036 | 0.311 | 0.009 |
| | | | post | 173.17 | 175.72 | | | | | |
| English | intervention | average adherence | pre | 117.7 | 132.52 | 1.09 | 396 | 0.082 | 0.775 | 0 |
| | | | post | 118.79 | 128.53 | | | | | |
| English | repetition | average no adherence | Pre | 436.65 | 277.48 | 0.12 | 114 | 0 | 0.991 | 0 |
| | | | post | 436.77 | 278.86 | | | | | |
| English | intervention | average no adherence | pre | 438.51 | 293.27 | 26.57 | 1100 | 61.024 | 0.000 | 0.053 |
| | | | post | 465.08 | 303.91 | | | | | |
| Hebrew | repetition | targeted adherence | pre | 227.88 | 272.33 | −29.94 | 143 | 3.434 | 0.066 | 0.024 |
| | | | post | 197.94 | 239.24 | | | | | |
| Hebrew | intervention | targeted adherence | pre | 192.13 | 248.97 | −11.24 | 585 | 3.432 | 0.064 | 0.006 |
| | | | post | 180.89 | 226.36 | | | | | |
| Hebrew | repetition | targeted no adherence | pre | 352.21 | 324.8 | 52.69 | 143 | 9.648 | 0.002 | 0.064 |
| | | | post | 404.9 | 310.28 | | | | | |
| Hebrew | intervention | targeted no adherence | pre | 327.74 | 312.6 | 97.74 | 585 | 125.99 | 0.000 | 0.177 |
| | | | post | 425.48 | 329.67 | | | | | |
| Hebrew | repetition | average adherence | pre | 162.49 | 179.83 | 3.22 | 143 | 0.174 | 0.678 | 0.001 |
| | | | post | 165.71 | 176.55 | | | | | |
| Hebrew | intervention | average adherence | pre | 117.31 | 154.67 | 19.31 | 250 | 18.921 | 0.000 | 0.071 |
| | | | post | 136.62 | 172.14 | | | | | |

(Continued.)

| language | condition | dependent variable | time | mean | s.d. | mean dif. | n | F | p-value | $\eta_p^2$ |
|---|---|---|---|---|---|---|---|---|---|---|
| Hebrew | repetition | average no adherence | pre | 467.18 | 293.65 | −1.22 | 143 | 0.018 | 0.895 | 0 |
| | | | post | 465.96 | 293.31 | | | | | |
| Hebrew | intervention | average no adherence | pre | 446.31 | 299.08 | 25.17 | 585 | 27.53 | 0.000 | 0.045 |
| | | | post | 471.48 | 313.14 | | | | | |
| Arabic | repetition | targeted adherence | pre | 272.26 | 320.78 | −61.79 | 137 | 7.66 | 0.006 | 0.053 |
| | | | post | 210.47 | 249.75 | | | | | |
| Arabic | intervention | targeted adherence | pre | 205.67 | 264.82 | −44.01 | 360 | 17.583 | 0.000 | 0.047 |
| | | | post | 161.66 | 227.33 | | | | | |
| Arabic | repetition | targeted no adherence | pre | 431.72 | 356.12 | 56.76 | 137 | 6.022 | 0.015 | 0.042 |
| | | | post | 488.48 | 352.31 | | | | | |
| Arabic | intervention | targeted no adherence | pre | 378.47 | 331.2 | 118.08 | 360 | 87.879 | 0.000 | 0.197 |
| | | | post | 496.55 | 349.14 | | | | | |
| Arabic | repetition | average adherence | pre | 181.84 | 186.17 | −12.2 | 137 | 1.325 | 0.252 | 0.01 |
| | | | post | 169.64 | 182.62 | | | | | |
| Arabic | intervention | average adherence | pre | 89.42 | 117.45 | −3.58 | 177 | 0.336 | 0.563 | 0.002 |
| | | | post | 85.84 | 109.45 | | | | | |
| Arabic | repetition | average no adherence | pre | 545.74 | 295.81 | 7.3 | 137 | 0.346 | 0.557 | 0.003 |
| | | | post | 553.04 | 310.52 | | | | | |
| Arabic | intervention | average no adherence | pre | 512.74 | 312.26 | 20.12 | 360 | 6.883 | 0.009 | 0.019 |
| | | | post | 532.86 | 320.45 | | | | | |
| Polish | repetition | targeted adherence | pre | 227.45 | 220.04 | −30.48 | 71 | 3.738 | 0.057 | 0.051 |
| | | | post | 196.97 | 182.28 | | | | | |

(Continued.)

| language | condition | dependent variable | time | mean | s.d. | n | mean dif. | F | p-value | $\eta_p^2$ |
|---|---|---|---|---|---|---|---|---|---|---|
| Polish | intervention | targeted adherence | pre | 254.7 | 257.02 | 395 | −29.22 | 14.733 | 0.000 | 0.036 |
| | | | post | 225.48 | 229.63 | | | | | |
| Polish | repetition | targeted no adherence | pre | 422.82 | 295.18 | 71 | 40.72 | 3.349 | 0.071 | 0.046 |
| | | | post | 463.54 | 271.87 | | | | | |
| Polish | intervention | targeted no adherence | pre | 424.86 | 319.71 | 395 | 75.7 | 65.106 | 0.000 | 0.142 |
| | | | post | 500.56 | 315.24 | | | | | |
| Polish | repetition | average adherence | pre | 173.87 | 157.49 | 71 | −3.59 | 0.175 | 0.677 | 0.002 |
| | | | post | 170.28 | 149.88 | | | | | |
| Polish | intervention | average adherence | pre | 185.13 | 178.43 | 110 | −0.62 | 0.011 | 0.917 | 0 |
| | | | post | 184.51 | 170.74 | | | | | |
| Polish | repetition | average no adherence | pre | 536.39 | 265.01 | 71 | 0.87 | 0.005 | 0.942 | 0 |
| | | | post | 537.26 | 269.84 | | | | | |
| Polish | intervention | average no adherence | pre | 541.42 | 288.78 | 395 | 20.04 | 12.554 | 0.000 | 0.031 |
| | | | post | 561.46 | 290.99 | | | | | |
| German | intervention | targeted adherence | pre | 196.56 | 217.59 | 126 | 0.93 | 0.006 | 0.936 | 0 |
| | | | post | 197.49 | 201.89 | | | | | |
| German | intervention | targeted no adherence | pre | 361.39 | 272.312 | 126 | 109.58 | 43.755 | 0.000 | 0.259 |
| | | | post | 470.97 | 296.11 | | | | | |
| German | intervention | average adherence | pre | 165.05 | 155.02 | 126 | 19.9 | 9.297 | 0.003 | 0.069 |
| | | | post | 184.95 | 168.75 | | | | | |
| German | intervention | average no adherence | pre | 484.73 | 265.17 | 126 | 39.26 | 18.773 | 0.000 | 0.131 |
| | | | post | 523.99 | 276.91 | | | | | |

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
