## [Reviewer comments · Royal Society Open Science]

Review History

RSOS-201131.R0 (Original submission)

Review form: Reviewer 1

Is the manuscript scientifically sound in its present form?

Yes

Are the interpretations and conclusions justified by the results?

Yes

Is the language acceptable?

Yes

Do you have any ethical concerns with this paper?

No

Have you any concerns about statistical analyses in this paper?

No

Recommendation?

Accept with minor revision (please list in comments)

Comments to the Author(s)

As with my previous round of comments, this is generally a clear, useful and convincing piece of research. Also as before, the exception is the initial Chicken Game, which is unnecessary for a piece that already has a lot going on in terms of the number of different treatments in play, and unconvincing on its own terms. It is far from clear that individuals perceive their decisions on social distancing and the like as a problem of strategic interaction. The preference ordering of the outcomes which defines the game is also open to challenge, e.g. the claim that unilateral adherence leads to "an unpleasant situation".

Decision letter (RSOS-201131.R0)

Dear Dr Fischer

On behalf of the Editors, I am pleased to inform you that your Manuscript RSOS-201131 entitled "The behavioral challenge of the COVID-19 pandemic: Indirect Measurements and Personalized Attitude Changing Treatments (IMPACT)" has been accepted for publication in Royal Society Open Science subject to minor revision in accordance with the referee suggestions. Please find the referees' comments at the end of this email.

The reviewers and handling editors have recommended publication, but also suggest some minor revisions to your manuscript. Therefore, I invite you to respond to the comments and revise your manuscript.

- Ethics statement

- Data accessibility

<http://datadryad.org/submit?journalID=RSOS&manu=RSOS-201131>

- Competing interests

- Authors' contributions

- Acknowledgements

- Funding statement

Because the schedule for publication is very tight, it is a condition of publication that you submit the revised version of your manuscript before 05-Aug-2020. Please note that the revision deadline will expire at 00.00am on this date. If you do not think you will be able to meet this date please let me know immediately.

If your manuscript is newly submitted and subsequently accepted for publication, you will be asked to pay the article processing charge, unless you request a waiver and this is approved by Royal Society Publishing. You can find out more about the charges at <https://royalsocietypublishing.org/rsos/charges>. Should you have any queries, please contact opscience@royalsociety.org.

Kind regards,
Andrew Dunn
Royal Society Open Science Editorial Office
Royal Society Open Science
opscience@royalsociety.org

on behalf of Dr Christina Demski (Associate Editor) and Essi Viding (Subject Editor)
opscience@royalsociety.org

Associate Editor Comments to Author (Dr Christina Demski):

Associate Editor: 1

Comments to the Author:

One of the original reviewers has now commented on the revised manuscript and has recommended acceptance with minor revision. In particular, the reviewer is still not sure whether

the Chicken game logic is necessary for the development of the paper and whether the same argument could not be made more simply as well. This may be because the Chicken game is scarcely mentioned after the introduction so could be better integrated in later sections. I would therefore like to ask you to provide a clearer and more elaborated rationale to the editorial team and the reviewer as to why you would like to keep it. Alternative (or in addition) you may want to remove it or integrate its relevance beyond the introduction.

Reviewer comments to Author:

Reviewer: 1

Comments to the Author(s)

As with my previous round of comments, this is generally a clear, useful and convincing piece of research. Also as before, the exception is the initial Chicken Game, which is unnecessary for a piece that already has a lot going on in terms of the number of different treatments in play, and unconvincing on its own terms. It is far from clear that individuals perceive their decisions on social distancing and the like as a problem of strategic interaction. The preference ordering of the outcomes which defines the game is also open to challenge, e.g. the claim that unilateral adherence leads to "an unpleasant situation".

Author's Response to Decision Letter for (RSOS-201131.R0)

See Appendix A.

Decision letter (RSOS-201131.R1)

Dear Dr Fischer,

It is a pleasure to accept your manuscript entitled "The behavioral challenge of the COVID-19 pandemic: Indirect Measurements and Personalized Attitude Changing Treatments (IMPACT)" in its current form for publication in Royal Society Open Science.

COVID-19 rapid publication process: We are taking steps to expedite the publication of research relevant to the pandemic. If you wish, you can opt to have your paper published as soon as it is ready, rather than waiting for it to be published the scheduled Wednesday.

This means your paper will not be included in the weekly media round-up which the Society sends to journalists ahead of publication. However, it will appear in the COVID-19 Publishing Collection which journalists will be directed to each week (<https://royalsocietypublishing.org/topic/special-collections/novel-coronavirus-outbreak>)

If you wish to have your paper published immediately please notify production@royalsociety.org and press@royalsociety.org when you respond to this email.

Appendix A

Response to Referees

In response to comments from the reviewer and the editor, regarding the role and importance of the Chicken game, we introduced the following changes:

1) We explain that the Chicken game highlight two important aspects: (i) It suggests that the behavioral challenge of reducing the effective reproductive number R matches the motivations of the involved individuals. Hence individuals' behavior is not only the problem, but, given appropriate interventions, may actually become the solution. (ii) Since the Chicken game is a member of a category of games termed Similarity Sensitive Games, cooperative behavior may be induced simply by raising the perception of similarity with other players.

2) We improved the description of the Chicken game in the caption of figure 1, and added another reference describing the cooperative nature of the Chicken game (Kun, Á., Boza, G., & Scheuring, I. (2006). Asynchronous snowdrift game with synergistic effect as a model of cooperation. *Behavioral Ecology*, 17(4), 633-641).

We revised the explanation of the preference ordering of the outcomes, specifically the explanation of the unilateral own-adherence payoff, now described as “being constrained by keeping health regulations while the other player is not. This is indeed an imbalanced and unfair outcome. Nonetheless, knowing the other player does not adhere, makes self-adherence even more valuable (as the alternative is to switch to no-adherence and obtain an even lower payoff)”. We have also revised the following description, explaining that “Since the game is symmetric, the intersection of the two Maxi-min strategies (i.e.: mutual cooperation) may be regarded as the natural outcome of the game (Rapoport and Guyer, 1966). Importantly, the game is also a Similarity Sensitive Game (Fischer, 2009, 2012), hence the higher the perception of similarity with the opponent the more likely one is to adhere to health regulations.”

3) In the description of our interventions we clarify that “Subjective Expected Relative Similarity (SERS): shows that cooperation in Similarity Sensitive Games (SSGs) **such as the Chicken game**, does not only depend on the expected payoffs, but also on the extent of perceived strategic similarity with the other party.

4) We return to the role of the Chicken game in the discussion, specifically to the congruence between the perspective of the individual and the perspective of the group and its role as a Similarity Sensitive Game, a game where cooperation follows the parties' perception of similarity with each other. The text in the revised discussed explains that "Analysis of the structure of the social dilemma underlying the situation, specifically the choice between adherence and non-adherence, revealed a typical payoff structure of a Chicken game. This showed that *in theory* individuals' adherence to health regulations is congruent with the public goal of reducing the effective reproductive number R . Since the Chicken game is a Similarity Sensitive Game, it also motivated one of the interventions, namely the addressing of similarity with the opponent as a means towards increased cooperation and enhanced adherence to health regulations."

We hope these revisions provide a clearer and more elaborated rationale for the role of the Chicken game and its inclusion in the study. We strongly believe that grounding our study on a game theoretic model that is built around the notion that 'what you do affects my behavior and what I do affects yours' is important conceptually and strengths the paper. Hence our desire to maintain its inclusion.